# A Robust Phenotypic High-Throughput Antiviral Assay for the Discovery of Rabies Virus Inhibitors

**DOI:** 10.3390/v15122292

**Published:** 2023-11-23

**Authors:** Xinyu Wang, Winston Chiu, Hugo Klaassen, Arnaud Marchand, Patrick Chaltin, Johan Neyts, Dirk Jochmans

**Affiliations:** 1Rega Institute, Department of Microbiology, Immunology and Transplantation, KU Leuven, Herestraat 49 Box 1043, 3000 Leuven, Belgium; xinyu.wang@kuleuven.be (X.W.); winston.chiu@kuleuven.be (W.C.); 2Cistim Leuven vzw, Bioincubator 2, Gaston Geenslaan 2, 3001 Leuven, Belgium; hugo.klaassen@cistim.be (H.K.); arnaud.marchand@cistim.be (A.M.); patrick.chaltin@kuleuven.be (P.C.); 3Center for Drug Design and Discovery (CD3), KU Leuven R&D, 3000 Leuven, Belgium

**Keywords:** rabies virus, high-throughput screening, drug repurposing, antiviral therapy

## Abstract

Rabies virus (RABV) causes severe neurological symptoms in mammals. The disease is almost inevitably lethal as soon as clinical symptoms appear. The use of rabies immunoglobulins (RIG) and vaccination in post-exposure prophylaxis (PEP) can provide efficient protection, but many people do not receive this treatment due to its high cost and/or limited availability. Highly potent small molecule antivirals are urgently needed to treat patients once symptoms develop. In this paper, we report on the development of a high-throughput phenotypic antiviral screening assay based on the infection of BHK-21 cells with a fluorescent reporter virus and high content imaging readout. The assay was used to screen a repurposing library of 3681 drugs (all had been studied in phase 1 clinical trials). From this series, salinomycin was found to selectively inhibit viral replication by blocking infection at the entry stage. This shows that a high-throughput assay enables the screening of large compound libraries for the purposes of identifying inhibitors of RABV replication. These can then be optimized through medicinal chemistry efforts and further developed into urgently needed drugs for the treatment of symptomatic rabies.

## 1. Introduction

The rabies virus (RABV) is a negative-stranded, non-segmented RNA virus that belongs to the genus of *Lyssavirus* in the family of the Rhabdoviridae. It is estimated that at least 60,000 people die annually because of rabies [1]. Once symptoms develop, mortality is nearly 100%. Post-exposure prophylaxis (PEP), for example after a dog bite, consists of a rabies immune globulins (RIGs) injection in and around the bite wound, along with vaccination. PEP can prevent most cases of rabies if administrated in time. However, the extensive quality assurance and an intact cold chain hinders access to PEP in medically underserved areas where endemic rabies occurs, especially in dogs. In addition, current rabies vaccines do not protect against infections with phylogroup II lyssaviruses (comprising genotypes 2 and 3 viruses Lagos bat and Mokola) [2].

In the Milwaukee protocol, which is a clinical intervention to treat symptomatic rabies, therapeutic coma is induced, and the patient receives a mix of drugs aimed at blocking viral replication (RIG’s, neutralizing monoclonal antibodies, ribavirin, and amantadine). However, there is very little evidence that this protocol is effective [3]. To date, small-scale research efforts have been devoted to explore the in vitro and in vivo effect of known antiviral agents [4,5,6,7,8,9,10,11,12], but there is still no licensed antiviral for rabies treatment. Recently, in mouse infection studies, the influenza inhibitor Favipiravir (T-705) was reported to result in some protective activity in a PEP-context but only when used in combination with vaccination [13,14]. 

Consequently, there is a very urgent need for highly potent drugs that can be used to treat patients who present with the first symptoms of rabies and who will almost certainly die. Highly effective antiviral mediations are successfully being used in the treatment of infections caused by herpesviruses [15], HIV [16], HBV [17], and HCV [18]. Moreover, small molecule antivirals are available for combating influenza [19] and SARS-CoV-2 [20] infection. There is every reason to believe that the development of potent drugs against rabies is feasible. Importantly, such drug(s) must have the ability to readily pass the blood–brain barrier when administered intravenously. Many antiviral targets in the virus replication cycle have yet to be explored, and phenotypic antiviral screenings may allow for the identification of molecules that target previously unknown and essential steps of the replication cycle of viruses. For instance, daclatasvir, which is approved for the treatment of hepatitis C virus (HCV), was discovered through high-throughput screening and was shown to be an inhibitor of a new antiviral target, i.e., NS5A [21]. Another example is our recent work, which reported on ultrapotent pan-serotype dengue virus inhibitors that prevent the interaction of two viral proteins (NS3 and NS4b) [22,23]. Also, this class of molecules was identified after a phenotypic screening of a large library of small molecules. To ensure the identification of a sufficient number of high-quality antiviral hits, it is essential that the phenotypic screens can be carried out in a robust high-throughput screening (HTS) format. 

Here we describe the development and validation of such an HTS-assay, which is based on the infection of BHK-21 cells with a reporter RABV and the use of high content imaging. As a first exercise, we screened a repurposing library composed of 3681 compounds, all of which had at least been studied in phase I clinical studies. Salinomycin was identified as the sole molecule in this series that resulted in some appreciable inhibition of viral replication by acting at the entry step. Hence, serious efforts to develop potent drugs against rabies are very urgently needed. 

## 2. Materials and Methods

### 2.1. Cells, Viruses, and Library

BHK-21 cells (ATCC^®^ CCL-10™) were cultured in Dulbecco’s Modified Eagle Medium (DMEM) (ThermoFisher Scientific, Brussels, Belgium) supplemented with 10% fetal bovine serum (FBS) (Hyclone, VWR, Leuven, Belgium) and penicillin-streptomycin (P/S) (100 U/mL, ThermoFisher Scientific). For RABV antiviral testing, cells were cultured in 2% FBS DMEM. SH-SY5Y cells (ATCC^®^ CRL-2266™) were grown in an DMEM/F-12 GlutaMAX™ Supplement (ThermoFisher Scientific) with 10% FBS, which was reduced to 2% FBS during antiviral testing. Rabies virus mCherry-SAD-B19 (1.6 × 10^6^ TCID_50_/mL) was kindly provided by Dr. Ashley C. Banyard (Animal & Plant Health Agency, London, UK) [24]. The mCherry sequence was inserted before the first gene within the genome of the SAD B19 strain. The RABV CVS-11 (9 × 10^5^ TCID_50_/mL) strain was kindly provided by Prof. Steven van Gucht (Sciensano, Ukkel, Belgium). The repurposing compound library (total of 3681 compounds) is composed of marketed and withdrawn, clinical compounds (Phase I to IV) as well as some well characterized and annotated biological tools. Library assembling was done by the Centre for Drug Design and Discovery (CD3, Leuven, Belgium). Each compound was dissolved in DMSO and was spotted with a 60 nL/well in 384-well plates; the final concentration of each compound in the assay was 10 µM. 

### 2.2. Optimized Protocol for 384-Well HTS Antiviral Assay and Image Analysis

Pre-spotted 384-well compound plates (containing 60 nL DMSO w/o compound in each well) were seeded with 2000 BHK-21 cells in a 30 µL medium per well and incubated for 24 h prior to the virus infection. On the following day, 30 µL of mCherry-RABV was added using non-contact liquid handlers at a multiplicity of infection (MOI) of 0.019 TCID_50_/cell and incubated for 4 additional days at 37 °C and 5% CO_2_. After incubation, Hoechst 33,342 (ThermoFisher Scientific) was added at a final concentration of 5 µM to stain nuclei for high content imaging (HCI) analysis.

High content imaging and analysis was performed using Arrayscan XTI (ThermoFisher Scientific). Images were collected by autofocusing on the nuclei stained by Hoechst using an excitation of 386 ± 23 nm, and the virus was captured in a secondary channel using an excitation of 560 ± 25 nm. Using a 5× objective, one field of view (covering 90% of the well) was selected to ensure sufficient image feature extraction and maintain scanning speed. Image analysis was performed with the corresponding HCS Studio software V6.6.2 (ThermoFisher Scientific), where a custom image analysis protocol was developed for this assay [25]. Briefly, separate nuclei were selected using a fixed pixel intensity threshold for object identification. Cells were defined as blue circles, as illustrated in the lower left image of Figure 1A. The mCherry reporter RABV was detected in the second channel using another fixed pixel intensity threshold. As demonstrated in the lower middle image of Figure 1A, infected cells were defined as green circles, while uninfected cells were denoted as orange circles. The infectivity percentage of each well was assessed by the ratio of the number of cells containing mCherry fluorescent signal (the green circles) to the total number of cells (the blue circles). The assessment of cell viability involved counting all separate nuclei. For wells containing compounds, cell viability was determined by normalizing the count to the infected controls. Since virus infection causes cell apoptosis, we considered the infected controls, instead of uninfected controls, to have 100% cell viability. The uninfected, untreated wells served as a quality control for the cells. Hit candidates were identified via selection of a cell viability greater than 60% and a virus infection less than 25%. These criteria allowed for conditions where the majority of the cells (>60%) survived compound exposure in the presence of the virus, and at the same time, only a minimal amount (<25%) of virus infection was observed.

### 2.3. Antiviral Assay for Hit Confirmation

In a previously developed antiviral assay [25], the fluorescent viral signal was quantified with a standard plate reader in 96-well plates. Similarly, in this study, hit compounds were serially diluted 2-fold, starting from 50 µM to yield 8 concentrations. A total of 15,000 BHK-21 cells were incubated with compounds overnight at 37 °C and 5% CO_2_ in a 150 µL assay medium. After the incubation, mCherry-SAD-B19 or CVS-11 strains were added at an MOI of 0.01 TCID_50_/cell in a volume of 50 µL and further incubated at 37 °C for 5 days. For the CVS-11 strain infection, the cells were fixed with 80% cold acetone for 30 min at 5 dpi. Following a 3-time PBS washing, RABV N protein was stained using a 1:100 diluted FITC Anti-Rabies Monoclonal Globulin (Fujirebio, Gent, Belgium). The fluorescence intensity from antibody staining (CVS-11) or mCherry expression by the virus (mCherry-SAD-B19) was quantified using a microplate reader (SPARK, Tecan, Mechelen, Belgium). Cell viability was determined by an MTS assay as described previously [26]. The program GraphPad Prism 9.0 was used to calculate the half maximal effective concentration (EC_50_), the 50% cytotoxic concentration (CC_50_), and the selectivity index (SI = CC_50_/EC_50_).

### 2.4. Time of Drug Addition Assay

BHK-21 cells were seeded into 24-well plates 24 h before the virus infection and compound treatment. On the following day, the cells were infected with RABV CVS-11 at an MOI of 1 TCID_50_/cell. Salinomycin (final concentration of 0.5 µM) was added at different time points (−1, 0, 0.5, 1, 2, 4 hpi). To determine whether salinomycin inhibits viral adsorption and attachment, the cells were incubated with the RABV CVS-11 strain at 4 °C for 1 h (−1–0 h) with or without a compound. To investigate whether salinomycin inhibits viral entry after adsorption and attachment, the unattached virus was washed away 3 times with cold PBS, cells were placed at 37 °C, and salinomycin was added at different time points (0, 0.5, 1, 2, 4 hpi). Infected cells without the compound treatment were defined as untreated controls, and untreated controls collected at 1 hpi were considered the virus background. At 16 hpi, intracellular RNA was extracted (E.Z.N.A. Total RNA Kit I R6834-01, Omega BIO-TEK, VWR, Leuven, Belgium) and quantified via RT-qPCR analysis.

### 2.5. Real-Time RT-qPCR

The rabies virus N gene was amplified with real-time quantitative PCR using iTaq™ Universal SYBR^®^ Green One-Step Kit (BIO-RAD, Temse, Belgium). The primers are as follows: 5′-AATGCGACGGTTATTGCTGC-3′ (forward) and 5′-TGCCACGTCGGACTTTGTTA-3′ (reverse). The standard curve was generated from the RNA extraction of the 1/10 diluted virus stock (9 × 10^5^ TCID_50_/mL). A 20 μL qPCR reaction contained 4 μL extracted sample RNA or standard, 10 μL iTaq Universal SYBR^®^ Green reaction mix, 0.25 μL reverse transcriptase, and 600 nM of each forward and reverse primer. qPCR was performed in a QuantStudio 5 real-time PCR system (Thermo Fisher Scientific) with the following procedure: 10 min at 50 °C for reverse transcription, 1 min at 95 °C for polymerase activation and DNA denaturation, and 40 cycles of 95 °C for 15 s and 62 °C for 30 s for amplification. Viral copies were calculated based on a standard curve using control material. 

### 2.6. Statistics

Statistical analysis in this study was done using GraphPad Prism 9. Dunnett’s multiple comparisons test was used to calculate the statistical significance. A *p*-value less than 0.05 was considered statistically significant. *p* values associated with each graph are indicated as follows: *, *p* value < 0.05; **, *p* value < 0.01; ***, *p* value < 0.001; ****, *p* value < 0.0001.

## 3. Results

### 3.1. Optimization of a High-Throughput Phenotypic Anti-RABV Assay

The RABV SAD B19 strain that was engineered to express the mCherry reporter (mCherry-SAD-B19) [26] was used to infect BHK-21 cells. At endpoint, the cell nuclei were stained with Hoechst, and, through image analysis, both the number of infected cells and the total cell count were quantified using a high-content imaging (HCI) protocol (Figure 1A). These parameters were subsequently used to calculate cell viability and the percentage of infected cells. When RABV-infected cells were left in culture for an extended period of several days, a cytopathic effect was observed, leading to a loss in the mCherry signal. Hence, the assay was first optimized in a 384-well plate format (2000 cells/well) by selecting the optimal viral inoculum and the optimal incubation time. At different time points after infection, cell viability (normalized to uninfected control) and the percentage of infected cells was determined (Figure 1B). When an inoculum of 0.019 TCID_50_/cell was selected and the cultures were incubated for 4 days, approximately 86% of the cells showed infection while viability was 71%, resulting in a Z’ of 0.88 (Figure 1C). This condition is considered optimal, as it allows for a limited virus inoculum and multiple rounds of virus replication with good signal reproducibility. 

### 3.2. Antiviral Screening of a Repurposing Library

A repurposing library of 3681 compounds was screened using the optimized 384-well HTS anti-RABV assay. A schematic screening protocol is shown in Figure 2A. Each compound was pre-spotted (60 nL/well of a 10 mM DMSO solution) in 384-well plates, and control wells received the same volume of DMSO without any compound. BHK-21 cells were seeded at 2000/well in a 30 µL medium and, 24 h later, 30 µL of virus inoculum at an MOI of 0.019 TCID_50_/cell was added. This resulted in a final compound concentration of 10 µM (0.1% (*v*/*v*) DMSO). On day 4 pi, imaging and analysis was performed, the percentage of infection was calculated for each well, and the percentage viability was calculated via normalization to the infected controls. Uninfected controls served as the quality control of the cell culture. Hits were identified as compounds with >60% of cell viability and <25% of virus infection as compared to the infected controls (Figure 3). This screening resulted in 4 primary hits and thus a ~0.1% hit rate. 

### 3.3. Validation of the Hit Compounds

To further validate the identified hits, their antiviral effect on mCherry-SAD-B19 virus replication in BHK-21 cells was measured in an assay whereby whole-well fluorescence was quantified using a standard plate reader (Figure 2B) [26]. The effect of the molecules on cell viability was assessed in uninfected BHK-21 cells cultured in 96-well plates, using the MTS viability staining method as described previously [27]. A schematic comparison of the 96-well assay and the 384-well HTS assay, together with an analysis of the readout, is shown in Appendix A. The Z’ factor and coefficient of variation (CV %) of the different assays are comparable (Appendix A). The signal-to-background ratio (S/B) is markedly better for the 384-well HCI assay but is still sufficiently high for the 96-well assay (Appendix A). Ribavirin resulted in comparable antiviral activity in both (96 or 384-well) assays (Appendix A). The confirmation assay was performed at concentrations ranging from 0.4 to 50 µM. Of the 4 primary hits, 2 have EC_50_ < 10 µM and CC_50_ > 10 µM. Only one molecule, salinomycin, was further investigated, as it had promising selective antiviral activity (selectivity index SI > 38) (Appendix A).

### 3.4. Salinomycin Inhibits RABV at an Early Stage of the Replication Cycle

Salinomycin (Figure 4A) is a monovalent ionophore, which is endowed with antibacterial, antiparasitic, and antitumoral activity [28]. To further validate the antiviral activity of the molecule and to determine if this activity is specific to the cell line or virus strain, it was also tested in SH-SY5Y cells (a cloned subline of the human neuroblastoma cell line SK-N-SH) and also against the laboratory-fixed CVS-11 strain. Salinomycin was somewhat less effective in the SH-SY5Y cell than in the BHK-21 cell (Figure 4B–E). The EC_50_s of salinomycin against SAD B19 strain are 0.051 µM in BHK-21 and 0.90 µM in SH-SY5Y cell; for the CVS-11 strain, the corresponding values are 0.18 µM and 0.67 µM, respectively. The CC_50_ is 4.0 µM both in BHK-21 and SH-SY5Y cells (Table 1). To investigate which step of the viral lifecycle is blocked by salinomycin, a time-of-drug-addition assay was performed. The drug had (at 0.5 µM) no inhibitory effect when it was added during the binding process at 4 °C (between −1 to 0 hpi) but resulted in a significant effect when added during the entry process at 37 °C (between 0 to 1 hpi). When added later in the replication cycle (at 2 to 16 hpi), no inhibitory effect was observed (Figure 4F,G). This indicates that salinomycin targets an early stage of RABV infection after the binding step.

## 4. Discussion

Despite the fact that rabies is almost inevitably lethal once neurological symptoms develop, there are no antiviral drugs available. Since the only treatment option of post-exposure prophylaxis (PEP) is costly and resources are limited, people are actively working on finding sustainable solutions. With the rapid development of automatic devices and computational technology, the search for target compounds of the rabies disease by various high-throughput screening (HTS) strategies is being conducted. Wu et al. identified clofazimine as a membrane fusion inhibitor using a rabies pseudovirus screening system [29]. Another entry inhibitor was identified from a single-cycle RABV reporter virus screening system [30]. However, there are limitations with the use of a pseudovirus in HTS, as only a part of the viral infection cycle can be studied. To contribute to the search for much needed antiviral drug(s), we developed a phenotypic antiviral assay that is amenable to high-throughput screening. To that end, we employed the fully replication competent RABV SAD B19 strain carrying a fluorescent reporter. The assay runs over several days to allow multiple rounds of rabies virus replication, ensuring that all potential antiviral targets are explored. Images of viral infection and cell nuclei are acquired using a high-content imaging (HCI) system, and an image analysis protocol allows for the automated evaluation of antiviral activity and cytotoxicity. This 384-well HTS anti-RABV assay is highly robust and exhibits excellent reproducibility.

Drug repurposing is a strategy used to discover novel applications for approved or investigational drugs that extend beyond their original medical indications. It offers advantages, including mitigating the risk of failure, reducing time frame and investment for drug development, and potentially uncovering new targets or mechanisms of treatment [31]. In this study, four molecules that inhibit RABV in vitro were identified from 3681 compounds. Cyproheptadines are used for migraine or psychotic disorders treatment; levamlodipine is a calcium channel blocker that is used for antihypertension and antianginal; dasatinib is a tyrosine kinase inhibitor used for the treatment of chronic myelogenous leukemia (CML) and acute lymphoblastic leukemia (ALL); and salinomycin is an inhibitor of Gram-positive bacteria. Due to salinomycin exhibiting a particularly high SI and a low EC_50_ in cell culture, it was selected for further investigation while the characterization of the other three confirmed hits is ongoing. Salinomycin is isolated from *Streptomyces albus* (strain no. 80614) [28]; it belongs to the group of polyether ionophore antibiotics, and it has been shown to have antibacterial (Gram-positive bacteria [32]), antifungal (filamentous fungi [33]), antiviral (influenza virus [34], SARS-CoV-2 [35]) and anticancer (human prostate cancer cell [35], angiogenesis effect [28]) activity. The in vitro anti-rabies activity of salinomycin was confirmed on different cell lines with different RABV strains, and preliminary experiments show that it targets an early stage of RABV infection after the virus binding step. This result is consistent with earlier reports indicating that it also inhibits the entry of influenza virus and SARS-CoV-2 [34,35]. As a polyether ionophore antibiotic, salinomycin is characterized as a transporter of monovalent and divalent cations (Na^+^, K^+^, Ca^2+^, Rb^+^, etc.) [36]. This is the mechanism by which salinomycin shows broad activity against virus, bacteria, and cancer cells, but this mechanism also results in severe neurotoxicity in vivo and in vitro. Studies indicate that salinomycin induces neurotoxic effects characterized by myelin loss, primary axonal degeneration and Wallerian-like degeneration in dogs, and distal polyneuropathy and peripheral nerve injury in cats [36]. Boehmerle et al. showed that salinomycin has a potential neurotoxic effect on human dorsal root ganglia and Schwann cells [37]. To date, the toxicity of salinomycin in the human neural system is still unknown. There has only been one reported human case of an accidental inhalation of salinomycin, which causes leg weakness, nausea, photophobia, and rhabdomyolysis [38]. More investigation is needed regarding the neurotoxicity of salinomycin treatment for humans.

In conclusion, this study developed a high-throughput screening assay suitable for antiviral compound discovery against RABV. A repurposing library was tested using an optimized screening assay. Salinomycin was identified as an entry inhibitor of RABV infection with low EC_50_ and good selectivity in cell culture. This study provides a new method of high-throughput RABV antiviral screening that is suitable for antiviral drug discovery.

## Figures and Tables

**Figure 1 viruses-15-02292-f001:**
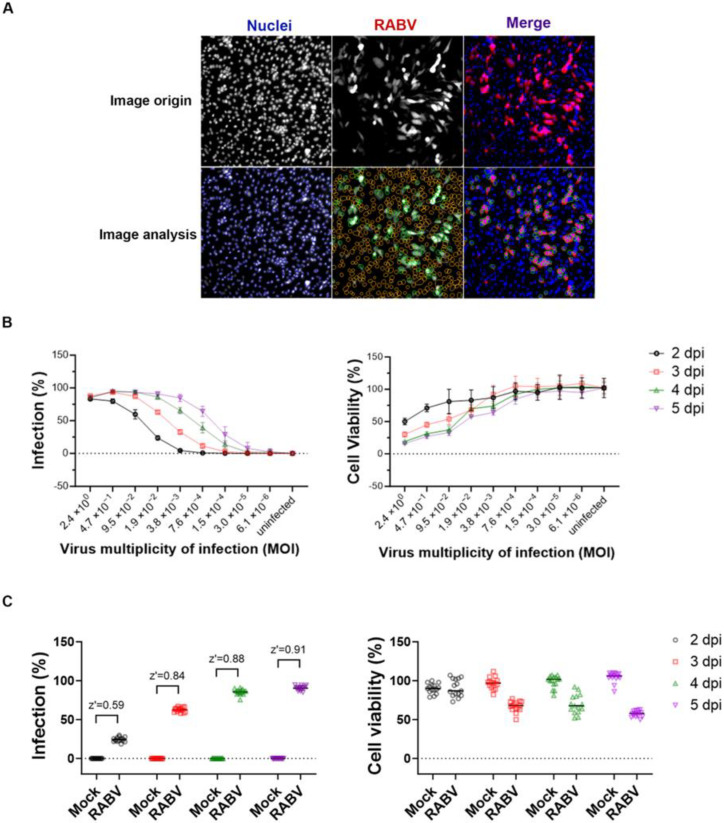
Optimization of the RABV 384-well antiviral screening. (**A**) Representative photomicrographs at 4 days after the infection of BHK-21 cells with RABV-mCherry. Unprocessed images of cell nuclei (Hoechst staining), RABV-mCherry, and the merger of both channels are depicted in the upper panels. For image analysis (bottom panels), the areas defined by a fixed pixel intensity threshold were selected and, based on nuclei staining, the cells were encircled with blue lines and counted as objects. Similarly, virus infection is detected in green circles; an orange circle within an object is counted as an uninfected cell. (**B**) Optimization of RABV dilutions and incubation time. Viral infection increased and viable cell number (nuclei count compared with uninfected controls) decreased with time. At an MOI of 0.019 TCID_50_/cell and a readout on day 4, an optimal condition was obtained with 86% infected cells and 71% cell viability. Data are from 3 independent experiments, and mean and standard errors are presented. (**C**) Based on the same dataset, the value of the Z’ factor and the percentage of cell viability at a MOI 0.019 TCID_50_/cell at different time points of readout are calculated.

**Figure 2 viruses-15-02292-f002:**
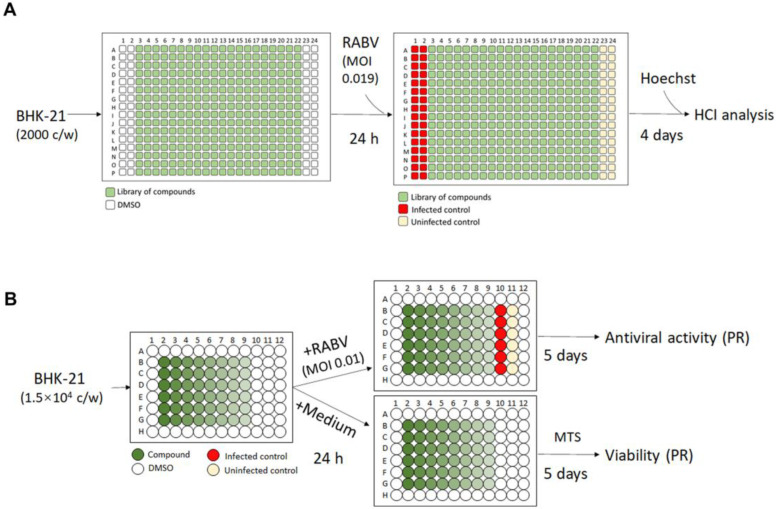
Schematic representation of the 384-well anti-RABV assay with high content imaging (HCI) used for HTS and the 96-well anti-RABV assay with whole-well fluorescence readout with a plate reader (PR) used for hit confirmation. (**A**) Schematic representation of the 384-well assay with HCI. After a 24 h incubation of 2000 BHK-21 cells with the test compounds, the cells were infected with mCherry-RABV (MOI 0.019 TCID_50_/cell). At 4 dpi, Hoechst was added to each well; virus infection and cell viability were assessed via HCI analysis. (**B**) Schematic representation of the 96-well assay. Test compounds were serially diluted and incubated with 1.5 × 10^4^ BHK-21 cells per well in 96-well plates. The next day, cultures were infected with mCherry-RABV (MOI 0.01 TCID_50_/cell), or the medium without the virus (viability assay) was added. On day 5 pi, the fluorescence intensity of the virus infection and cell viability (MTS assay) were quantified (using a plate reader (PR)).

**Figure 3 viruses-15-02292-f003:**
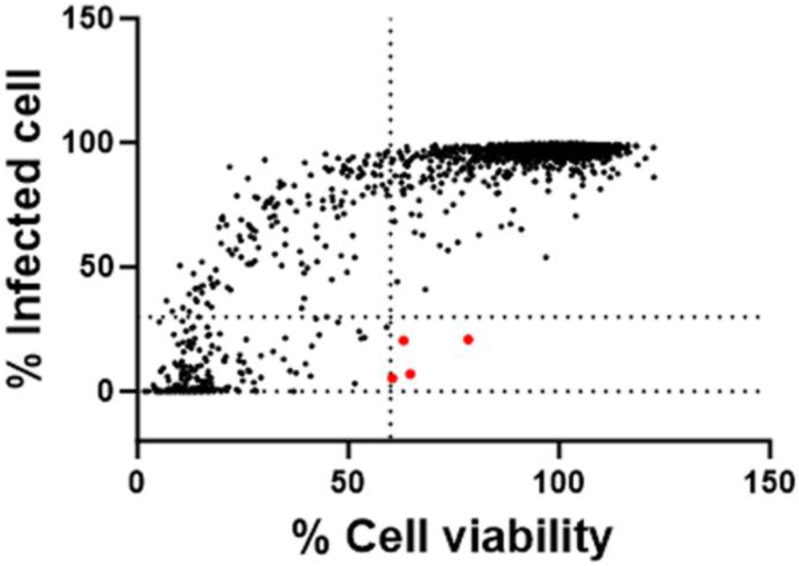
Results of the antiviral screen of a repurposing library of 3681 compounds. Each dot represents the result of the testing of a single compound at 10 µM. The area with red dots indicates the selection criteria for the hits (>60% cell viability, <25% virus infection), resulting in the identification of 4 hits.

**Figure 4 viruses-15-02292-f004:**
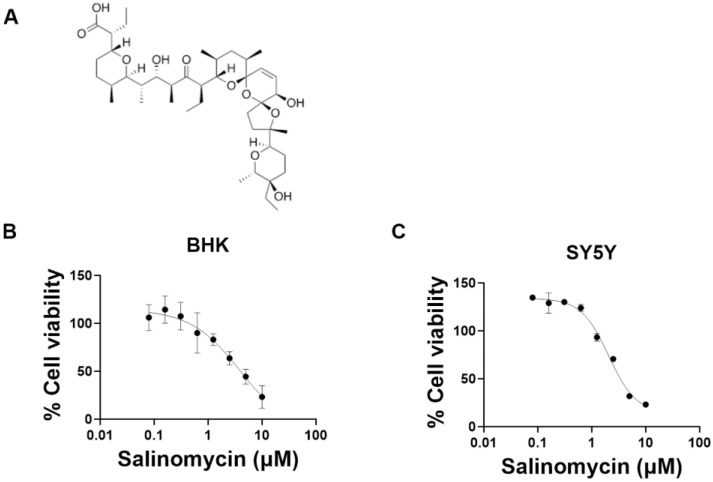
Salinomycin inhibits RABV at an early stage of infection. (**A**) Chemical structure of salinomycin. (**B**–**E**) Dose response effect of salinomycin on RABV replication and on the viability of BHK-21 and SH-SY5Y cells. Each condition was tested in three independent assays; averages and STDEV are indicated. Calculated EC_50_s in BHK-21 cells are 0.051 µM and 0.18 µM on SAD B19 and CVS-11, respectively; and 0.9 µM and 0.67 µM on SAD B19 and CVS-11, respectively, in SH-SY5Y cells. Calculated CC_50_s are 4.0 µM on BHK-21 and SH-SY5Y cells. (**F**) Schematic of the time-of-drug-addition assay. BHK-21 cells were infected with RABV (CVS-11) either with or without the compound and incubated at 4 °C for 1 h (−1 to 0 hpi). After 3 washes with cold PBS, the plate was transferred to 37 °C (at 0 hpi). Salinomycin (0.5 µM) was added at various time points (−1, 0, 1, 0.5, 1, 2, 4 hpi) before collecting the cells at 16 hpi. Infected cells without the compound treatment were defined as untreated controls, and the sample that was collected at 1 hpi was considered the input virus (background). (**G**) Intracellular viral RNA of samples from different time points. Hydroxychloroquine (HOQ) was used as a reference compound for the virus entry. Data were from three independent assays; averages and STDEV are given. Dunnett’s multiple comparisons test was used to calculate the statistical significance. (* *p* < 0.05; ** *p*< 0.01; *** *p* < 0.001).

**Table 1 viruses-15-02292-t001:** Antiviral activity of salinomycin in vitro.

Virus Strain	^1^ EC_50_ (µM)		CC_50_ (µM)	
	BHK-21	SH-SY5Y	BHK-21	SH-SY5Y
SAD B19	0.051 ± 0.020	0.90 ± 0.070	4.0 ± 0.18	4.0 ± 0.060
CVS-11	0.18 ± 0.0040	0.67 ± 0.11		

^1^ EC_50_ and CC_50_ are the average and SD of three independent experiments.

## Data Availability

Data is contained within the article or Appendix A.

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
