# Peer review of "A Robust Phenotypic High-Throughput Antiviral Assay for the Discovery of Rabies Virus Inhibitors"

_viruses, 2023, doi:10.3390/v15122292_

Round 1

Reviewer 1 Report

Comments and Suggestions for Authors

The search for original molecules to suppress the reproduction of the rabies virus is certainly relevant. The manuscript contains an original description of a laboratory assey for screening antiviral agents. The method allows to obtain representative and reproducible results, the processing of which can be automated. The results of screening 3681 compounds are presented and the best result (salinomycin) is described in more detail. The results obtained by the authors can be used for further study of the inhibitory effect of salinomycin on the rabies virus in vivo.

A few notes on the manuscript.

1.       In different places in the text, different values are given to indicate the criteria (cell vialability and virus infection) for selecting compounds. For example:

line108:   50% and 30%

lines 196-197:  60% and 25%

Figure 3:  50% and 25%

Besides it is not clear on what basis these criteria were chosen (quartiles?). It would be useful to note this when describing methods.

2.        Also in different places in the text, different numbers of compounds that were selected as a result of the study are mentioned. In Figure 3, 4 red dots are visible (probably 4 compounds), the same number is indicated in the text (line 197). The discussion mentions 10 compounds, of which 6 show a dose-responsed effect (line 289). Table S 1 includes 5 compounds. Only salinomycin is listed in the abstract, but results for other molecules (Table S1) are also of interest.

Author Response

We higly appreciate the comments of reviewer 1 and are gratefull for the time invested. We provide our awnsers  below each topic raised by the reviewer:

The search for original molecules to suppress the reproduction of the rabies virus is certainly relevant. The manuscript contains an original description of a laboratory assey for screening antiviral agents. The method allows to obtain representative and reproducible results, the processing of which can be automated. The results of screening 3681 compounds are presented and the best result (salinomycin) is described in more detail. The results obtained by the authors can be used for further study of the inhibitory effect of salinomycin on the rabies virus in vivo.

 A few notes on the manuscript.

  1. In different places in the text, different values are given to indicate the criteria (cell vialability and virus infection) for selecting compounds. For example:

line108:   50% and 30%

lines 196-197:  60% and 25%

Figure 3:  50% and 25%

Besides it is not clear on what basis these criteria were chosen (quartiles?). It would be useful to note this when describing methods.

We thank the reviewer for pointing out these discrepancies. The selection criteria of >60% viability and <25% virus infection was used in this study. All discrepancies are now corrected. Since compounds in this screening are tested at a single concentration, these hit-selection criteria are chosen based on our experience from previous screenings.

We included this in line 119-122 “These criteria allow for conditions where the majority of the cells (>60%) survived compound exposure in the presence of virus, and at the same time only a minimal amount (<25%) of virus infection is observed.”

  1. Also in different places in the text, different numbers of compounds that were selected as a result of the study are mentioned. In Figure 3, 4 red dots are visible (probably 4 compounds), the same number is indicated in the text (line 197). The discussion mentions 10 compounds, of which 6 show a dose-responsed effect (line 289). Table S 1 includes 5 compounds. Only salinomycin is listed in the abstract, but results for other molecules (Table S1) are also of interest.

We thank the reviewer for pointing out this anomaly. In earlier analysis different selection criteria were used which resulted in this inconsistency in the manuscript. With the final criteria (>60% viability and <25% virus infection), 4 compounds are defined as hits, and this is now consistently presented in the manuscript. The discussion has rephrased to: “In this study, 4 molecules that inhibit RABV in vitro were identified from 3681 compounds.” in lines 305-306.

The four compounds (Cyproheptadine, Levamlodipine, Dasatinib and Salinomycin) in Table S1 are 4 hits, ribavirin is shown as a reference compound – this information is now included in the legend of the table. As mentioned in the discussion, due to salinomycin exhibiting a particularly high SI (>10) and a low EC50 in cell culture, it was selected for further investigation. While other compounds may undergo further investigation, they are not the focus of this study.

Reviewer 2 Report

Comments and Suggestions for Authors

The manuscript by Wang and colleagues is essentially a methods paper describing a high throughput assay to identify antiviral compounds that have activity against rabies virus.  With small exceptions, the manuscript is well written and logically presented.  Certainly, effective antivirals targeting rabies virus is an important need and the assay system described could well contribute to first pass screening of large compound libraries.  In that sense, this manuscript should be of interest to the field, and may be applicable to other pathogens.  I do not have major criticisms but offer the following minor suggestions:

First line of abstract: sever -> severe

Second line: disease is almost inevitably lethal

Line 30: suggest rural areas -> in underserved areas where endemic rabies occurs, especially in dogs.

Line 47: also -> development of

Line 49: druggable is, I think, slang.  Suggest changing to susceptible.

Line 157: carry -> express the

Comments on the Quality of English Language

Very minor suggestions

Author Response

We also higly appreciate the comments of reviewer 2 and are gratefull for the time he/she invested. We provide our awnsers  below each topic raised by the reviewer:

The manuscript by Wang and colleagues is essentially a methods paper describing a high throughput assay to identify antiviral compounds that have activity against rabies virus.  With small exceptions, the manuscript is well written and logically presented.  Certainly, effective antivirals targeting rabies virus is an important need and the assay system described could well contribute to first pass screening of large compound libraries.  In that sense, this manuscript should be of interest to the field, and may be applicable to other pathogens.  I do not have major criticisms but offer the following minor suggestions:

We thank the reviewer for these suggestions to improve the text and made changes accordingly.

  1. First line of abstract: sever -> severe
    We corrected the word “sever” to “severe” in line 8.
  2. Second line: disease is almost inevitably lethal
    We corrected “uniformly” to “almost inevitably” in line 9 and line 283
  3. Line 30: suggest rural areas -> in underserved areas where endemic rabies occurs, especially in dogs.
    We rephrased the sentence in line 29-31 as: “However, the extensive quality assurance and an intact cold-chain hinders access to PEP in medically underserved areas where endemic rabies occurs, especially in dogs.”
  4. Line 47: also -> development of
    We changed the word “also” to “development of” in line 48.
  5. Line 49: druggable is, I think, slang.  Suggest changing to susceptible.
    We deleted the word “druggable” in this sentence in line 50.
  6. Line 157: carry -> express the
    We changed the word “carry” to “express the” in line 169.

Reviewer 3 Report

Comments and Suggestions for Authors

In this manuscript, Wang and co-authors describe a small molecule phenotypic high-throughput screen against RABV. The authors screened a chemical compounds library, composed of 3681 clinical stage compounds, identifying Salinomycin as selective entry inhibitor.

The work represents a good proof of concept, amenable to future screening of larger libraries to address an unmet clinical need for RABV antiviral treatments.

The study is well-controlled, and the results are clearly presented overall, although a discrepancy in the list of hits listed in the results vs the discussion is noticed.

 Below are my suggestions for improvement.

1)      In paragraph 3.2 of the results and in Figure 3 it is indicated that 4 hits were identified from the primary screen, while in the discussion it is mentioned that 10 molecules were identified as hits, of which 6 confirmed in dose-response. Chlorprothixene, Saracatinib and CCT128930 are only cited in the discussion, and not mentioned in the results and/or figures. Please, explain this discrepancy. In addition, there is an inconsistency in the selection criteria for hit calling in the M&M session (>50% viability, <30% virus infection) vs the results and Figure 3 (>60% viability, <25% virus infection). Please, clarify which cutoffs were used.

2)      In lines 235-236 it is stated “In either cell line salinomycin was somewhat less effective against the CVS-11 than the SAD B19 strain”. In SH-SY5Y cells salinomycin is more potent against CVS-11 (0.76 uM) than against SAD B19 (0.9 uM). Please, edit the sentence to better reflect the results.

3)      Lines 52-54: Daclatasvir should be included as relevant example of an approved drug identified through HTS approach (DOI: 10.1038/nature08960)

4)      From Figure 3, it looks like both the cell count and the antiviral data have been normalized. However, no specification is included in the text about the normalization of the % of infection. Please, specify if those data have also been normalized to the DMSO-treated infected wells.

Author Response

We also higly appreciate the comments of reviewer 3 and are again gratefull for the time he/she invested. We provide our awnsers  below each topic raised by the reviewer:

In this manuscript, Wang and co-authors describe a small molecule phenotypic high-throughput screen against RABV. The authors screened a chemical compounds library, composed of 3681 clinical stage compounds, identifying Salinomycin as selective entry inhibitor.

The work represents a good proof of concept, amenable to future screening of larger libraries to address an unmet clinical need for RABV antiviral treatments.

The study is well-controlled, and the results are clearly presented overall, although a discrepancy in the list of hits listed in the results vs the discussion is noticed.

 Below are my suggestions for improvement.

1)  In paragraph 3.2 of the results and in Figure 3 it is indicated that 4 hits were identified from the primary screen, while in the discussion it is mentioned that 10 molecules were identified as hits, of which 6 confirmed in dose-response. Chlorprothixene, Saracatinib and CCT128930 are only cited in the discussion, and not mentioned in the results and/or figures. Please, explain this discrepancy. In addition, there is an inconsistency in the selection criteria for hit calling in the M&M session (>50% viability, <30% virus infection) vs the results and Figure 3 (>60% viability, <25% virus infection). Please, clarify which cutoffs were used.

We thank the reviewer for pointing out this anomaly. In earlier analysis different selection criteria were used which resulted in this inconsistency in the manuscript. With the final criteria (>60% viability and <25% virus infection), 4 compounds are defined as hits, and this is now consistently presented in the manuscript. Chlorprothixene, Saracatinib and CCT128930 are therefore not mentioned in the manuscript anymore. Of note: these compounds were retested but no antiviral activity could be confirmed.

We corrected the sentence in M&M in lines 118-119 to: “Hit candidates were identified by the selection for a cell viability greater than 60% and virus infection less than 25%.”

Changes in discussion in lines 305-312 were made: “In this study, 4 molecules that inhibit RABV in vitro were identified from 3681 compounds. Cyproheptadine is used for migraine or psychotic disorders treatment; levamlodipine is a calcium channel blocker used for antihypertension and antianginal; dasatinib is a tyrosine kinase inhibitor used for the treatment of chronic myelogenous leukemia (CML) and acute lymphoblastic leukemia (ALL); and salinomycin is an inhibitor of gram-positive bacteria.”

2) In lines 235-236 it is stated “In either cell line salinomycin was somewhat less effective against the CVS-11 than the SAD B19 strain”. In SH-SY5Y cells salinomycin is more potent against CVS-11 (0.76 uM) than against SAD B19 (0.9 uM). Please, edit the sentence to better reflect the results.

The mentioned sentence was changed to: “Salinomycin was somewhat less effective in SH-SY5Y cell than in BHK-21 cell (Figure 4B, C, D, E).” in lines 250-252.

3) Lines 52-54: Daclatasvir should be included as relevant example of an approved drug identified through HTS approach (DOI: 10.1038/nature08960)

We thank the reviewer for this suggestion. Information on Daclatasvir was added in the introduction section in lines 53-54: “For instance, daclatasvir, which is approved for the treatment of hepatitis C virus (HCV), was discovered through high-throughput screening and was shown to be an inhibitor of a new antiviral target i.e. NS5A [21].”

4) From Figure 3, it looks like both the cell count and the antiviral data have been normalized. However, no specification is included in the text about the normalization of the % of infection. Please, specify if those data have also been normalized to the DMSO-treated infected wells.

More specific interpretation of percentage of infection was changed in M&M section in lines 102-109: “Briefly, separate nuclei were selected using a fixed pixel intensity threshold for object identification. Cells were defined as blue circles, as illustrated in the lower-left image of Figure 1A. The mCherry reporter RABV was detected in the second channel using another fixed pixel intensity threshold. As demonstrated in the lower-middle image of Figure 1A, infected cells were defined as green circles, while uninfected cells were denoted as orange circles. The infectivity percentage of each well was assessed by the ratio of the number of cells containing mCherry fluorescent signal (the green circles) to the total number of cells (the blue circles).”